# *Aconitum* Alkaloid Songorine Exerts Potent Gamma-Aminobutyric Acid-A Receptor Agonist Action In Vivo and Effectively Decreases Anxiety without Adverse Sedative or Psychomotor Effects in the Rat

**DOI:** 10.3390/pharmaceutics14102067

**Published:** 2022-09-28

**Authors:** Zsolt Kristóf Bali, Nóra Bruszt, Zsombor Kőszegi, Lili Veronika Nagy, Tamás Atlasz, Péter Kovács, Dezső Csupor, Boglárka Csupor-Löffler, István Hernádi

**Affiliations:** 1Grastyán Endre Translational Research Centre, University of Pécs, 6 Ifjúság Str., H-7624 Pécs, Hungary; 2Translational Neuroscience Research Group, Centre for Neuroscience, Szentágothai Research Centre, University of Pécs, 20 Ifjúság Str., H-7624 Pécs, Hungary; 3Institute of Physiology, Medical School, University of Pécs, 12 Szigeti Str., H-7624 Pécs, Hungary; 4Department of Experimental Zoology and Neurobiology, University of Pécs, 6 Ifjúság Str., H-7624 Pécs, Hungary; 5Department of Pharmacognosy, University of Szeged, 6 Eötvös Str., H-6720 Szeged, Hungary; 6Institute for Translational Medicine, Medical School, University of Pécs, 12 Szigeti Str., H-7624 Pécs, Hungary; 7Institute of Clinical Pharmacy, University of Szeged, 8 Szikra Str., H-6720 Szeged, Hungary

**Keywords:** GABA-A receptors, in vivo electrophysiology, microiontophoresis, vigilance, anxiety, behavioral pharmacology, diterpene alkaloids, picrotoxin, saclofen, songorine

## Abstract

Songorine (SON) is a diterpenoid alkaloid from *Aconitum* plants. Preparations of Aconitum roots have been employed in traditional oriental herbal medicine, however, their mechanisms of action are still unclear. Since GABA-receptors are possible brain targets of SON, we investigated which subtypes of GABA-receptors contribute to the effects of SON, and how SON affects anxiety-like trait behavior and psychomotor cognitive performance of rats. First, we investigated the effects of microiontophoretically applied SON alone and combined with GABA-receptor agents picrotoxin and saclofen on neuronal firing activity in various brain areas. Next, putative anxiolytic effects of SON (1.0–3.0 mg/kg) were tested against the GABA-receptor positive allosteric modulator reference compound diazepam (1.0–5.0 mg/kg) in the elevated zero maze (EOM). Furthermore, basic cognitive effects were assessed in a rodent version of the psychomotor vigilance task (PVT). Local application of SON predominantly inhibited the firing activity of neurons. This inhibitory effect of SON was successfully blocked by GABA(A)-receptor antagonist picrotoxin but not by GABA(B)-receptor antagonist saclofen. Similar to GABA(A)-receptor positive allosteric modulator diazepam, SON increased the time spent by animals in the open quadrants of the EOM without any signs of adverse psychomotor and cognitive effects observed in the PVT. We showed that, under in vivo conditions, SON acts as a potent GABA(A)-receptor agonist and effectively decreases anxiety without observable side effects. The present findings facilitate the deeper understanding of the mechanism of action and the widespread pharmacological use of diterpene alkaloids in various CNS indications.

## 1. Introduction

For hundreds of years, extracts of various species of the *Aconitum* genus (Ranunculaceae) have been used in traditional Chinese and Japanese medicine. They are applied for several purposes, e.g., as analgesics, or antirheumatics [1], however, several poisoning cases have been reported with *Aconitum* extracts [2] due to their strong toxicity. Therefore, pure formulations of certain bioactive components of the *Aconitum* plants may be more useful in pharmacotherapy. To date, more than 450 diterpenoid and non-diterpenoid alkaloids have been isolated from *Aconitum* species [3,4].

Songorine (SON) is a typical diterpenoid alkaloid which was first isolated from the plant *Aconitum soongaricum*. The therapeutic potential of SON has been implicated in several applications including cardiac arrhythmia, inflammatory diseases, and wound healing [3,5]. The diverse peripheral effects of SON may imply a wide range of cellular targets and mechanisms including the induction of nuclear factor-erythroid factor 2-related factor 2 (Nrf2) and the inhibition of membrane Ca^2+^ channels, growth factor receptors, and cytokine expression [6,7,8,9]. Pharmacokinetic studies have confirmed that systemically applied SON rapidly distributes in body tissues, and passes the blood–brain barrier resulting in a well-detectable level of the drug in the CNS [10,11]. Regarding the effects of SON on higher-order CNS functions, only sparse data are available so far which suggest the potential anxiolytic and even procognitive action of SON in laboratory mice [12,13] However, only a few studies have investigated the psychopharmacological targets of SON, and the findings are not yet coherent. An earlier in vitro study suggested that SON may act as a putative gamma-aminobutyric acid (GABA) receptor A antagonist based on electrophysiological recordings in hippocampal slices of young rats [14] without any in vivo evidence for the similar action of SON. However, the earlier-mentioned putative anxiolytic activity of SON contradicts the GABA(A) receptor antagonist effect. Therefore, the present study was designed to further investigate the neuronal effects of SON in the rodent brain under in vivo conditions and, for the first time, to examine the cellular-level interaction of SON with known GABAergic receptor ligands. In addition, we also aimed to extend the existing evidence by confirming or rejecting the current notions on putative behavioral effects of SON. Therefore, we measured anxiety-like behavior of rats in a standardized elevated zero maze (EOM) behavioral paradigm [15]. Furthermore, we aimed to test whether SON exerts negative (sedative) effects through reducing alertness and vigilance, known as generally accepted side-effects of GABAergic anxiolytic agents. For this purpose, we designed and validated a rat version of the psychomotor vigilance task (PVT) [16,17]. As an ultimate aim, we set out to clarify the presently contradicting interpretations of earlier findings and facilitate the understanding of the basic neurochemical grounds of the effects of SON in the living mammalian brain.

## 2. Materials and Methods

### 2.1. Animal Care

Animals were kept in conventional animal houses of the University of Pécs under a 12/12 h light/dark cycle with controlled temperature and humidity. Animals used for electrophysiology were ad libitum fed with standard laboratory chow, whereas animals for behavioral experiments were fed with 17 g/animal of laboratory chow to ensure their sufficient exploratory drive in the behavioral tests. Water was available ad libitum to all animals. Experiments were approved by the Animal Care Committee of the University of Pécs, and the Department of Animal Health and Food Control of the County Government Offices of the Ministry of Agriculture. Measures were taken to minimize pain and discomfort of the animals in accordance with the Directive 40/2013 (II.14): “On animal experiments” issued by the Government of Hungary, and the Directive 2010/63/EU “On the protection of animals used for scientific purposes” issued by the European Parliament and the European Council.

### 2.2. In Vivo Electrophysiology

#### 2.2.1. Surgery

Altogether, 31 adult male Long–Evans rats (Charles River Laboratories, Gödöllő, Hungary) were examined during the electrophysiological experiments. Anesthesia was induced with a single injection of ketamine (100 mg/kg, SBH, Budapest, Hungary) and was monitored by performing the tail-flick test regularly. Additional low doses of ketamine were administered, when necessary, to maintain stable levels of anesthesia. After opening the skull, the dura mater was incised, and a recording microelectrode was lowered into the brain tissue with a hydraulic micromanipulator (Narishige, Tokyo, Japan). The recording electrode was positioned at the coordinates of AP: −3.6 mm, ML: 2.6 mm from Bregma and V: 0.5–7 mm from dura according to Paxinos and Watson (1997) [18]. From recording tracks at these coordinates we were able to sample neuronal activity from various forebrain structures including the cerebral neocortex and archicortex and the diencephalon.

#### 2.2.2. Extracellular Recordings

Seven-barrel microelectrodes (Carbostar-7S, Kation Scientific, Minneapolis, MN, USA) were used for recording the single-unit activity via the central carbon fiber of the electrode. Extracellular single-unit activity was amplified and filtered using a biological amplifier (Supertech Ltd., Pécs, Hungary). Analog data were passed through an analog–digital converter (Power 1401, CED, Cambridge, UK) to a PC. Recording, spike sorting, and data analysis were performed using Spike2 software (CED, Cambridge, UK). Frequency histograms were built, and the neuronal activity (firing rate) was expressed as cycles per second (Hz).

#### 2.2.3. Microiontophoretic Drug Delivery

Microiontophoretic application of pharmacological compounds was carried out through the micropipettes (microcapillaries) surrounding the central carbon fiber of the seven-barrel microelectrode [19,20]. One of the pipettes of the electrode, which was filled with saline solution (0.9% NaCl), was used for the application of a continuous balancing current with opposite polarity, while other pipettes were filled with one of the following bioactive substances (abbreviation, pipette concentration, and vehicle in parentheses): songorine (SON, 16 mM, in 33% DMSO), picrotoxin (PIC, 5 mM, in distilled water), saclofen (SAC, 33 mM, in distilled water), and DMSO (33% in distilled water). Gamma-aminobutyric acid (GABA, 200 mM, in distilled water) was used as positive control for inhibitory effects. Control applications of DMSO confirmed that the vehicle of SON did not affect neuronal firing.

Picrotoxin, SAC, DMSO, and GABA were purchased from Sigma-Aldrich (St. Louis, MO, USA), whereas SON was extracted from the roots of *Aconitum toxicum Reichenb.* (Ranunculaceae) at the Department of Pharmacognosy, University of Szeged, Hungary according to a previously validated protocol by Csupor et al., 2006 [21]. Compounds were ejected by applying positive (SON, PIC, GABA) or negative (SAC) current (typically between 0–100 nA) on the pipettes using individual constant-current circuits (Neurophore BH2, Medical Systems Corp., Greenvale, NY, USA). 

#### 2.2.4. Data Analysis and Statistics

Pharmacological effects on the neural activity were expressed as normalized firing rate, i.e., the ratio of the firing rate after and before microiontophoretic drug application. One sample *t*-test with a reference value of 1.00 was applied to assess the statistical significance of the firing rate decreasing or increasing effect of the test compounds. Normalized firing rate values were also used for comparing effects and efficacies of different test compounds using one-way ANOVA followed by post-hoc Tukey’s HSD test.

Comparison of the single effect of SON to the co-application of GABA receptor antagonist compounds (i.e., PIC or SAC) was made by comparing the distributions of the number of neurons that responded with firing rate increase, decrease, or no change to the treatments using contingency tables and chi-squared test. A neuron was considered responsive to a treatment (i.e., showing firing rate increase or decrease) if its firing rate was significantly different in a 10s time-window before and after the administration of a compound (Student’s *t*-test), otherwise the neuron was assigned to the not-responding category.

In all hypothesis tests, statistical significance was accepted at *p* < 0.05.

### 2.3. Elevated Zero Maze (EOM) Test

#### 2.3.1. Initial Group Assignment

Two to six days before the EOM experiments, rats were tested in the open field (OF) apparatus to generate experimental groups for the EOM test that showed uniform locomotor activity. For the open field test, a 57.5 cm × 57.5 cm (length × width) sized dark grey plywood box was used. Each rat spent 3 min in the OF apparatus. The locomotor activity of rats was assessed as the number of crossings through the thin black lines painted on the floor of the apparatus (four by four lines in equal distances from each other). From the initial group of rats, four were excluded from the forthcoming EOM test because of outlier performance in the OF test. Then, anxiolytic effects of SON were tested in the EOM apparatus.

#### 2.3.2. Apparatus

The EOM apparatus (Maze Engineers, Glenview, IL, USA) consisted of a circular platform placed on 61 cm tall stands. The diameter of the maze was 100 cm, the width of the platform was 10 cm and was divided to four quarters. Two quarters of the platform were enclosed with walls 30 cm in height (closed quarters) while two quarters had no walls (open quarters). The two closed quarters were separated from each other by the two open quarters. The length of closed and open platforms was the same.

#### 2.3.3. Test Protocol

At the beginning of the EOM experiment, the rat was placed in one of the closed quarters, and its behavior was observed for 5 min. During the session, time spent in open quarters, the number of visits in the open quarters, the number of head-dippings, and the number of rearings in the closed quarters were registered. Head-dipping was defined as the event when the animal looks down over the edge of the open platform by placing its head under the level of the platform.

#### 2.3.4. Pharmacological Treatments and Experimental Design

In the first EOM experiment, 48 naïve adult (6–7 months old) male Long–Evans rats (Janvier Labs, Le Genest-Saint-Isle, France; *n* = 12 per group) were treated with different doses (1.0, 2.5, and 5.0 mg/kg b.w.) of the reference anxiolytic compound diazepam (DZP) for validation of the testing apparatus and the paradigm used. Diazepam was obtained as a liquid pharmaceutical formulation (Seduxen, Gedeon Richter Pharmaceutical Co., Budapest, Hungary), and was diluted using a vehicle similar in composition to the original vehicle of Seduxen, containing 50 mg/mL sodium benzoate dissolved in the mixture of 40% propylene glycol, 10% ethanol, and 50% sterile distilled water [22]. The same solution was injected in animals of the control group (VEH). Diazepam or the vehicle was injected subcutaneously (s.c.) in 1 mL/kg volume at 30 min before the beginning of the EOM test.

In the second EOM experiment, the effects of SON (in doses: 1.0, 2.0, and 3.0 mg/kg b.w.) were tested on the behavior of 36 naïve adult (6–7 months old) male Long–Evans rats (*n* = 9 per group) in the EOM. For the behavioral experiments, SON was purchased from BOC Sciences (Shirley, NY, USA) and was dissolved in 3.3% DMSO in phosphate-buffered saline (0.1 M, pH = 6.3). The same vehicle was used for preparing dilutions and for the injection of control rats (VEH). Songorine or its vehicle was injected s.c. in 2 mL/kg volume approximately 45 min before the beginning of the test (based on pilot observations).

#### 2.3.5. Data Analysis and Statistics

In the EOM test, pharmacological treatments with DZP and SON were tested in between-subject design against the corresponding VEH groups. The time spent in the open quarters was analyzed using univariate GLM in the SPSS software (v26; IBM, Armonk, NY, USA). If a significant main effect was found, groups that received treatments with the pharmacological agents (DZP or SON) were compared to the VEH group using Dunnett’s post-hoc test. As the number of visits in open quarters, the number of head-dippings, and the number of rearings were non-continuous variables, the non-parametric Kruskal–Wallis test was used to evaluate the effect of treatments on the given parameter. A rejected null hypothesis in the Kruskal–Wallis test was followed by Dunn’s post-hoc test with adjustment of *p*-values using Bonferroni’s method. In every statistical test, a *p*-value less than 0.05 was considered significant, while a *p*-value between 0.05 and 0.10 was considered as a tendency toward a significant result.

### 2.4. Psychomotor Vigilance Task

Since both DZP and SON supposedly exert marked anxiolytic effects measurable in the EOM task, it is reasonable to assess their putative effects on basic alertness and vigilance. Therefore, we further examined the psychopharmacological effects of DZP and SON in the PVT. In the present experiments, we designed a rodent version of the PVT which is widely used in human clinical studies and applied research for the measurement of basic motor and cognitive functions.

#### 2.4.1. Apparatus and Test Protocol

We used a standardized operant conditioning apparatus for rats (Habitest System, Coulbourn Instruments, Holliston, MA, USA), equipped with stimulus generators (feeder light, LED cue lights), response sensors (levers, photocell) and reward pellet delivery modules (feeder with delivery trough). Adult male Long–Evans rats were gradually trained in a 10-step procedure to fully perform the PVT.

Every trial of a PVT session was run as follows: Rats were required to respond to the onset of the centrally located feeder light module with nose-poking into the pellet delivery trough of the feeder module. Then, rats had to hold their nose in the delivery trough until the feeder light was turned off (fixation period). The fixation period varied randomly between 0 to 5000 ms. After the offset of the feeder light, the cue lights were illuminated above the two levers located left and right to the central feeder module, and rats had to pull their nose out of the pellet delivery trough and press one of the levers within 10 s. After the lever-press response, the rat received a reward pellet (45 mg dustless precision pellet, Bio-Serv, Flemington, NJ, USA), the cue lights were also turned off, and began an intertrial interval (ITI) of 15 s. If the animal accomplished the given trial, the total reaction time (RT) was measured as the time from the offset of the feeder light (onset of the cue lights) until the lever press response. If the trial was not correctly performed, different kinds of errors were defined and registered. If the animal did not put its nose in the feeder trough within 10 s after the onset of the feeder light, the trial was not initiated, and it was considered as a missed trial. If the animal nose-poked but removed its head from the feeder trough before the offset of the feeder light, a premature response was considered. If the animal succeeded the fixation period but did not press the lever after the onset of the cue lights, an omission error (lapse) was considered. All types of errors resulted in the offset of all light stimuli, and a 5 s punishment period followed by an ITI of 15 s. In each experimental session, rats performed the task for 60 min or until 90 initiated trials.

#### 2.4.2. Pharmacological Treatments and Experimental Design

Different doses of DZP (1.0, 2.5, and 5.0 mg/kg, s.c.) and SON (1.0, 2.0, and 3.0 mg/kg, s.c.) were tested against their vehicles (same as in the EOM experiments) in a within-subject design by applying different treatments in different experimental sessions (days) according to a counterbalanced Latin-square arrangement. Injection of DZP and SON were carried out in 8 and 12 rats 15 and 30 min before the start of each session, respectively. Drug administrations were carried out in the same volume as in the EOM experiments.

#### 2.4.3. Data Analysis and Statistics

The effects of DZP and SON on PVT performance was assessed by analyzing RT and the number of different kinds of errors. The total reaction time was split into two components: (1) the time elapsed from the offset of the feeder light until the removal of the head from the feeder trough was called “nose-out time”, and was mainly considered as the decision phase of the response; (2) the time elapsed from the removal of the head from the feeder trough until the lever-press response was called “lever-pressing time”, and was mainly considered as the motor execution phase of the response.

The results of PVT experiments were evaluated using a linear mixed-effects model with a random intercept for repeated measurements in IBM SPSS v26. If a significant main effect (*p* < 0.05) of the treatment was found on a given measure, effects of different doses of the drug were compared to the vehicle treatment using the least significant difference (LSD) post-hoc test.

## 3. Results

### 3.1. In Vivo Electrophysiology

Altogether, 433 neurons were extracellularly recorded from different brain areas (neocortex, hippocampus, and thalamus). Firing activity of three representative recordings are depicted in Figure 1. Control DMSO applications did not modify baseline neuronal activity. Effects of the test compounds on neuronal firing rate are summarized in Table 1 and in Figure 2. A single microiontophoretic injection of GABA caused remarkable suppression of the spontaneous neuronal activity (93/123 neurons, 75.6%). The grand average of the normalized firing rate as a result of GABA administration confirmed the strong inhibitory effects (0.604 ± 0.049 Hz, *p* < 0.001). We did not observe any latency in the effect of GABA, and at the end of the microiontophoretic application, the original firing rate recovered almost immediately. Single applications of SON caused significant inhibition in neuronal firing rate (90/112 neurons, 80.4%) in each tested brain area to an average normalized firing rate of 0.621 ± 0.041 Hz (*p* < 0.001). Firing-rate change was not immediately observable but occurred a few seconds after the beginning of SON application. The latency was dependent on the applied microiontophoretic current. Latency was shorter when a higher current was applied on the microiontophoretic channel of SON, and vice versa. In several cases, the inhibitory effect persisted after the end of SON application (for up to 10 min), and the original spontaneous firing rate slowly and gradually returned to the baseline. Dose-dependence calculations for SON and GABA showed similar functions of microiontophoretic current. The application of higher current caused significantly higher inhibition of the single unit activity in the case of both GABA and SON. According to ANOVA, there was no difference between the magnitude of the inhibitory effects of GABA and SON (F(5, 427) = 17.256, *p* < 0.001; GABA vs. SON: *p* > 0.05, Figure 2).

Neurons responded somewhat differently to the single application of GABA(A) receptor antagonist PIC and the GABA(B) receptor antagonist SAC. Picrotoxin increased the neuronal firing rate in 42.9% of neurons (24/56 neurons) with no neurons showing observable inhibitory responses. However, the normalized firing rate was not significantly higher than the control level (1.150 ± 0.091 Hz, *p* = 0.105). On the other hand, SAC exerted excitatory effect only in 22.5% of neurons (11/49 neurons), and only a few neurons were inhibited after the application of SAC (3/49 neurons). The average normalized firing rate during SAC application was not significantly higher than the baseline (1.066 ± 0.037 Hz, *p* = 0.106). According to ANOVA results, normalized firing rates during the administration of GABA receptor antagonists PIC and SAC were significantly higher than during the application of GABA and SON (F(5, 427) = 17.256, *p* < 0.001; GABA vs. PIC: *p* < 0.001, GABA vs. SAC: *p* < 0.001, SON vs. PIC: *p* < 0.001, SON vs. SAC: *p* < 0.001; Figure 2).

After investigating the effects of single applications of all test compounds, we tested the receptor selectivity of SON by performing co-applications of GABA agonists and antagonists. Inhibitory effects of SON were counteracted by GABA(A) receptor antagonist PIC but not by GABA(B) receptor antagonist SAC. When PIC was co-applied with SON, inhibitory effects of SON were observed only in case of 23.2% of neurons (13/56 neurons), and most neurons did not change the firing activity as a consequence of SON + PIC application (42/56 neurons). The distribution of neurons that showed firing rate increase, decrease, or no change was significantly different between the single application of SON and the co-application of SON and PIC (χ = 51.634, *p* < 0.001). Although a certain extent of firing rate decrease was still observed (0.915 ± 0.034 Hz, *p* = 0.016), the normalized firing rate during SON + PIC co-application showed the amelioration of the inhibitory effect of SON when co-applied with PIC (ANOVA: SON vs. SON + PIC: 0.621 ± 0.041 Hz vs. 0.915 ± 0.034 Hz, *p* < 0.001, Figure 2). Furthermore, the normalized firing rate during the co-application of SON + PIC was significantly lower than after the single application of PIC (PIC vs. SON + PIC: 1.150 ± 0.091 Hz vs. 0.915 ± 0.034 Hz, *p* < 0.01).

On the other hand, SAC did not change the percentage of neurons that were inhibited by SON: 81.1% of neurons (30/37 neurons) decreased their firing rate during SON + SAC co-application which is comparable to the inhibitory effects of SON on 80.4% of neurons during its single applications (χ = 3.252, *p* = 0.197). Normalized firing rate during SON + SAC was also similar to that during single SON application (SON vs. SON + SAC: 0.621 ± 0.041 Hz vs. 0.617 ± 0.089 Hz; *p* > 0.05) but lower than during the single application of SAC (SAC vs. SON + SAC: 1.069 ± 0.042 Hz vs. 0.617 ± 0.089 Hz; *p* < 0.001). Moreover, neurons significantly decreased their firing rate compared to the baseline (*p* < 0.001) which clearly shows that SAC did not block the inhibitory effects of SON. Furthermore, SAC did not affect the inhibitory action of SON even when a longer application of SAC was applied before the ejection of SON (See also Figure 1).

### 3.2. Assessment of the Anxiolytic Effects of SON

According to the currently observed GABA(A) receptor agonist-like effect of SON, we tested its possible anxiolytic effects in comparison with diazepam (DZP), a known anxiolytic GABA(A) receptor ligand. The behavior of rats after different treatments was assessed in the elevated zero maze (EOM) test.

The time spent in the open quarters was considered as the primary parameter of the EOM test, and DZP treatment showed a significant main effect in this measurement (F(3, 33) = 3.209, *n* = 38, *p* < 0.05, Figure 3). Diazepam at the 2.5 mg/kg dose significantly increased the time that the animals spent in the open quarters compared to the VEH treatment (DZP2.5 vs. VEH: 86.5 ± 19.8 s vs. 41.5 ± 5.4 s, *p* < 0.05). Thus, DZP at 2.5 mg/kg dose successfully decreased anxiety and provided a reliable positive control for anxiolytic effect in the EOM test. Secondary measurements of the number of open quarter visits and head-dippings were not affected by DZP treatment (H = 1.251, df = 3, *p* = 0.741; and H = 2.734, df = 3, *p* = 0.434, respectively). On the other hand, DZP dose-dependently decreased the number of rearings (measured in the closed quarters), exerting a significant decrease in vertical exploratory activity at 2.5 mg/kg and 5.0 mg/kg doses compared to the VEH treatment (H = 17.257, df = 3, *p* < 0.001; VEH: 5.9 ± 6.0 rearings, DZP2.5: 0.5 ± 0.5 rearings, DZP5.0: 1.5 ± 1.0 rearings; DZP2.5 vs. VEH: *p* < 0.01, DZP5.0 vs. VEH: *p* < 0.01).

Songorine showed a dose-dependent anxiolytic effect in the EOM (Figure 4) as the time spent in open quarters significantly increased with ascending doses of SON (F(3, 23) = 3.726, *n* = 27, *p* < 0.05). Songorine was found effective in decreasing anxiety-like behavior of rats at the 3.0 mg/kg dose as SON3.0 treatment significantly increased the time spent in the open quarters (SON3.0 vs. VEH: 83.4 ± 12.7 s vs. 25.1 ± 5.7 s, *p* < 0.05). Moreover, SON also increased the number of open quarter visits as a marginally significant effect of treatments was found (H = 7.493, df = 3, *p* = 0.058). Post-hoc comparisons revealed that the 3.0 mg/kg dose of SON also significantly increased the number of open-quarter visits (SON3.0 vs. VEH: 7.4 ± 0.4 vs. 2.7 ± 0.8, *p* < 0.05). This result further confirms the anxiolytic activity of SON. Although there was a slight increase in the number of head-dippings, this secondary parameter was not significantly affected by SON (H = 3.182, df = 3, *p* = 0.364). In contrast with DZP, SON did not affect the number of rearings in the closed quarters (H = 1.982, df = 3, *p* = 0.576), suggesting that SON does not express motor side-effects that were seen after DZP treatments.

### 3.3. Assessment of the Effects of SON in the Psychomotor Vigilance Task

Basic psychomotor and cognitive effects of DZP and SON were tested in the PVT task using 8 and 12 rats, respectively, in a within-subject experimental design (Figure 5). After treatment with DZP, the RT of rats in the PVT showed a tendency to slow down to the randomly introduced target stimuli. However, significant increase of RT was only detected in the lever-pressing component of the reaction time (F(3, 15.2) = 5.337, *p* < 0.05) and not in the earlier nose-out (decision) phase. Diazepam in a dose of 5.0 mg/kg significantly slowed down the late motor responses of the rats compared to the control measurements with vehicle treatment (lever-pressing: DZP5.0 vs. VEH: 1.13 ± 0.29 s vs. 0.71 ± 0.11 s, *p* < 0.01). Furthermore, DZP in both 2.5 and 5.0 mg/kg doses increased the number of omission errors, when rats failed to press the lever following the end of fixation period and the onset of cue lights (F(3, 17) = 3.116, *p* = 0.054; DZP2.5 vs. VEH: 6.17 ± 2.72 vs. 0.25 ± 0.16, *p* < 0.05; DZP5.0 vs. VEH: 7.50 ± 4.66 vs. 0.25 ± 0.16; *p* < 0.05). Diazepam also showed a tendency to decrease the overall activity of the rats in the PVT (F(3, 15.4) = 2.540, *p* = 0.095), since 5.0 mg/kg DZP increased the number of missed trials (DCZ5.0 vs. VEH: 49.0 ± 17.8 vs. 13.1 ± 7.5; *p* < 0.05). Together these data show that DZP markedly decreased the activity and speed of rats in the PVT and mainly affected the motor execution phase of the responses.

In contrast, treatment with SON did not induce any impairment in the performance of rats in the PVT in any of the applied doses. Songorine did not affect either reaction time (RT: F(3, 33) = 0.171, *p* = 0.915; nose-out: F(3, 33) = 0.116, *p* = 0.950; lever-pressing: F(3, 33) = 1.072, *p* = 0.374) or the number of errors (premature responses: F(3, 33) = 1.876, *p* = 0.153; omissions: F(3, 33) = 0.672, *p* = 0.575; missed trials: F(3, 33) = 0.530, *p* = 0.665). Therefore, PVT data fully support our observations in the EOM test that SON had no detrimental side-effects on basic psychomotor and cognitive functions of rats.

## 4. Discussion

The present study provides the first in vivo data about the potent inhibitory effect of SON on spontaneous neuronal firing activity of forebrain neurons and the data suggest that the inhibitory activity of SON is dependent on the activation of GABA(A) receptors. The presently reported electrophysiological evidence is in line with the observed behavioral pharmacological activity of SON, as, in the EOM experiments, we confirmed that SON is a potent anxiolytic compound that effectively decreased anxiety-like behavior in rats. The observed anxiolytic activity of SON was found similar to the GABA(A) positive allosteric modulator DZP. In addition, sedation and profound locomotor side-effects (typical for DZP) were not observed after SON treatment in the PVT. Songorine did not impair the speed of responses or the overall performance compared to DZP which markedly increased motor-execution time and number of omission errors. Thus, SON acted as a potent anxiolytic agent without showing any of the typical side effects of benzodiazepines [23].

There are only a few studies available that have investigated the CNS effects of SON. Zhao et al. (2003) described the specific binding of SON to GABA(A) receptors on synaptic membranes of rats [14]. The authors found that the inward currents elicited by GABA were inhibited by SON in whole-cell voltage-clamp experiments on neurons isolated from rat hippocampus in vitro. Thus, the authors suggested a GABA(A) receptor antagonist activity of SON. The latter conclusion is in contrast with our present findings showing that SON decreased firing activity of neurons in vivo, and this inhibitory action was readily blocked by the GABA(A)-receptor antagonist picrotoxin. Furthermore, the suggestion of Zhao et al. (2003) about a GABA(A) receptor antagonist effect of SON is also in conflict with earlier [12] and presently reported anxiolytic behavioral effects of SON. A possible cause of the conflicting results may be that Zhao et al. (2003) investigated neurons that were isolated from very young (P5 to P9 days old) rat pups [14]. Neuronal responses to GABA are known to change during development [24,25], thus, a reasonable explanation of these seemingly conflicting findings is that the applied drugs exert different effects in animals of different postnatal ages. Nevertheless, the affinity of SON to GABA(A) receptors has been concordantly confirmed by both Zhao et al. (2003) and our own results [14].

However, as is usual in the case of natural compounds, SON may have several pharmacological targets in the central nervous system. As the currently reported behavioral paradigms (EOM, PVT) may not selectively measure GABAergic effects, it is reasonable to suppose that binding of SON to other putative targets such as dopamine receptors might also contribute to its observed anxiolytic potential. In an in vitro electrophysiological study, Ameri (1998) [26] reported that SON exerted a stimulatory effect on synaptic transmission in the hippocampus by increasing field EPSPs in the dendritic area of CA1 pyramidal neurons and postsynaptic population spikes. However, as neither the presynaptic population spikes nor the antidromically evoked population spikes were modulated by the administration of SON, the author concluded that SON does not exert a general excitatory effect or presynaptic facilitation of neurotransmitter release. Rather, the observed stimulatory effects reported by Ameri (1998) [26] were attributed to the putative agonistic action of SON on dopaminergic receptors, since the increase of postsynaptic population spikes by SON was successfully blocked by the selective D2/D3 receptor antagonist sulpiride but not by the selective D1 receptor antagonist SCH23390.

Such possible dopaminergic action of SON in addition to its presently reported GABA(A)-receptor agonist effect seems to be reasonable from the perspective of our present behavioral findings and earlier data [12,13]. In line with an earlier experiment of Nesterova et al. (2015) [12], the animals in the present EOM experiment showed a marked decrease of anxiety-like behavior after the administration of SON similar to the GABA(A) receptor positive allosteric modulator DZP but without the signs of locomotor disturbances that are typical in the case of benzodiazepine derivatives [23]. Furthermore, we confirmed the lack of such side-effects on attention, vigilance, and psychomotor functions in the PVT, where neither the slowing of responses nor the increased numbers of the different types of errors were observed. As the PVT is the most widely used test battery for the assessment of alertness, vigilance, and certain aspects of higher-order (executive) functions and provides the best known translational potential between human [17,27], non-human primate [28], and rodent behaviors [29], we can conclude that SON, in the applied effective anxiolytic doses, does not induce measurable psychomotor and cognitive side-effects. The importance of the dopaminergic system is well-known in maintaining the necessary motivational levels and setting the optimal time-accuracy function underlying sustained attention and vigilance [30,31]. Therefore, it is reasonable to suppose that SON may exert its beneficial action, at least partially, on dopaminergic receptor terminals in the forebrain or on certain GABA-receptors of the midbrain that may consequently facilitate DA release from midbrain dopaminergic neurons [32]. Both mechanisms may facilitate the general anxiolytic effects and foster cognitive performance while preventing adverse locomotor side-effects. Further investigations will shed light on brain-region and neuron-type specific actions of SON and will reveal the interacting neuronal circuitries behind the presently observed anxiolytic effects.

Although the effectiveness of SON for several conditions has been shown in preclinical studies [3], so far, no clinical trials have investigated its potential benefits in human therapy. Obviously, the body of evidence of the pharmacokinetic and pharmacodynamic properties, as well as of the mechanisms of action of SON are still not sufficiently large to step further towards testing on humans. Hopefully, our present report and other relevant findings will promote further initiatives aiming at the better understanding of the pharmacological action of SON, the identification of cellular targets, and possibly, the design of several lead molecules by the modification of SON for further drug development.

In summary, our study presents the first in vivo electrophysiological evidence for the agonist action of SON at GABA(A) but not GABA(B) receptors. We further confirmed the potent anxiolytic action of SON in the widely accepted EOM test of anxiety-like behavior of rats and demonstrated the lack of psychomotor side-effects in the PVT. These findings together extend our knowledge of the pharmacological effects of SON in the CNS. The present results may also open further research avenues considering the possible widespread pharmacological potential of diterpene alkaloids in neurological and psychiatric conditions.

## Figures and Tables

**Figure 1 pharmaceutics-14-02067-f001:**
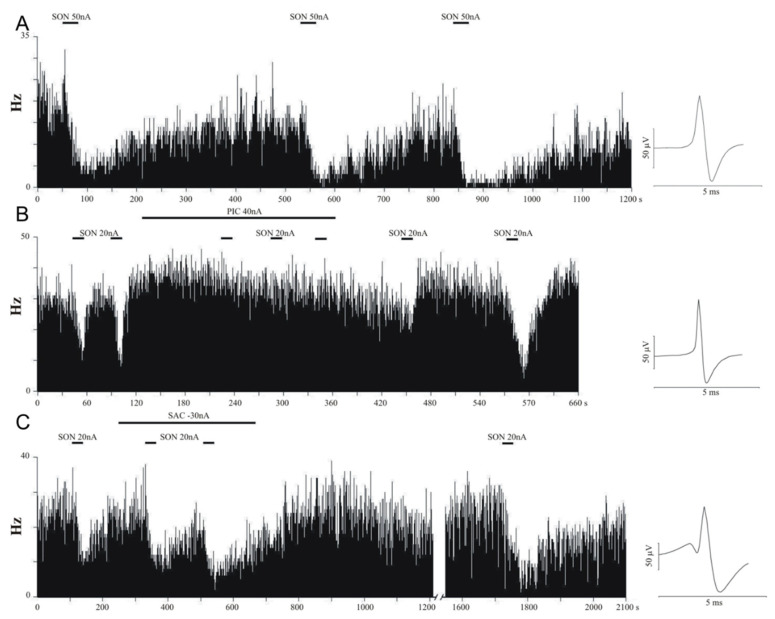
Frequency histograms of representative neurons from the rat forebrain. Microiontophoretic injection of songorine (SON) inhibited neuronal activity (**A**). After the termination of the ejection of SON, the spontaneous neuronal activity always returned. Picrotoxin (PIC) successfully blocked the inhibitory effects of SON (**B**). However, the application of saclofen (SAC) did not modulate the inhibitory effects of SON (**C**).

**Figure 2 pharmaceutics-14-02067-f002:**
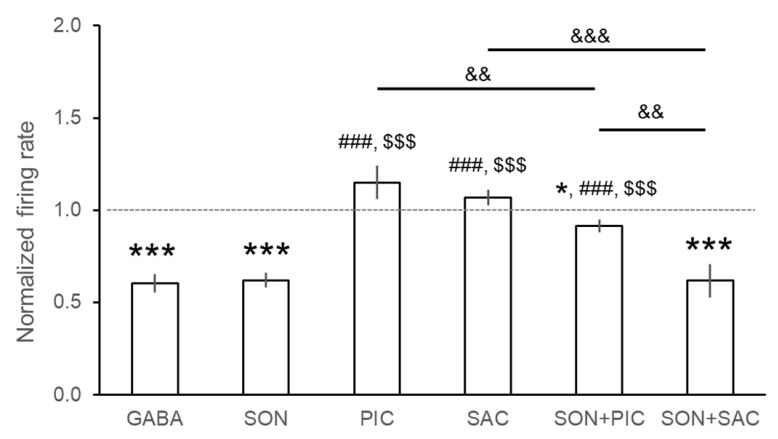
Effects of various treatments on neuronal firing activity. Injections of GABA or SON inhibited neuronal firing activity, while injections of PIC or SAC did not change the firing rate. The GABA(A) antagonist PIC was able to attenuate the inhibitory effects of SON. However, the GABA(B) antagonist SAC did not influence the modulatory effects of SON (one-way ANOVA: F(5, 427) = 17.256, *p* < 0.001). Notation of significant differences are as follows: one-sample *t*-test: * *p* < 0.05, *** *p* < 0.001 vs. control level. Post-hoc comparisons according to Tukey’s Q: ### *p* < 0.001 vs. SON, $$$ *p* < 0.001 vs. GABA, && *p* < 0.01, &&& *p* < 0.001 in comparisons between groups marked by horizontal lines.

**Figure 3 pharmaceutics-14-02067-f003:**
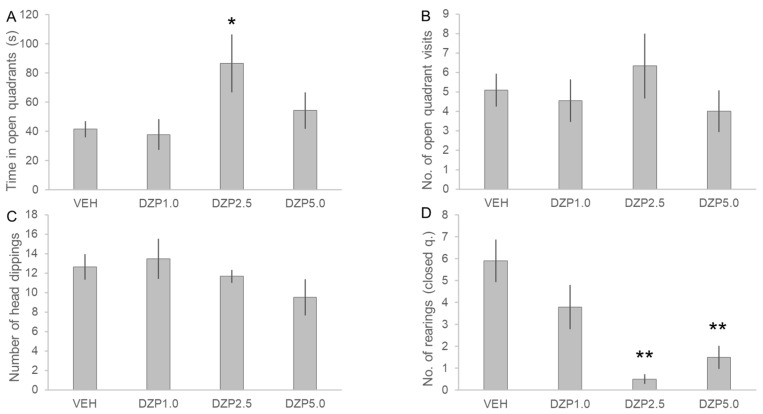
Effects of diazepam on the behavior of rats in the elevated zero maze (EOM) test. Level of anxiety was assessed by measuring the time spent in the open quadrants (**A**), the number of visits to the open quadrants (**B**), and the number of head-dippings (**C**). As an additional variable, the number of rearings in the closed quadrants (**D**) was also counted to assess pharmacological effects of the treatments on the locomotion of the rats. Data are expressed as mean ± SEM. VEH: vehicle, DZP1.0: 1.0 mg/kg diazepam, DZP2.5: 2.5 mg/kg diazepam, DZP5.0: 5.0 mg/kg diazepam. Asterisks mark significant effects of the given treatment compared to the VEH treatment according to post-hoc Dunnett’s test (**A**) or Dunn’s test (**B**–**D**): * *p* < 0.05, ** *p* < 0.01.

**Figure 4 pharmaceutics-14-02067-f004:**
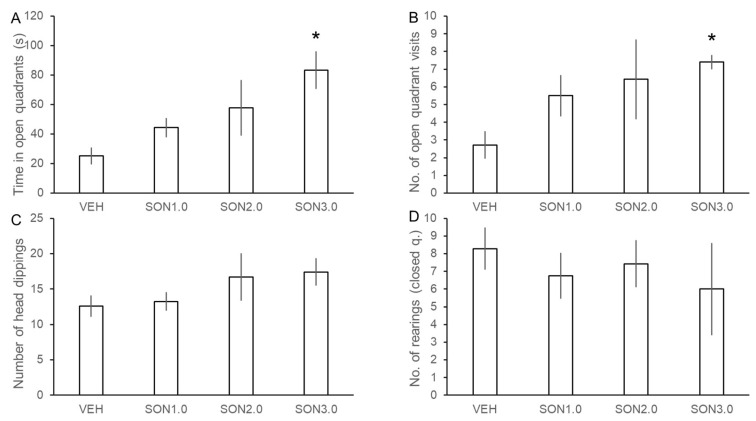
Effects of songorine on the behavior of rats in the elevated zero maze (EOM) test. Level of anxiety was assessed by measuring the time spent in the open quadrants (**A**), the number of visits to the open quadrants (**B**), and the number of head-dippings (**C**). As an additional variable, the number of rearings in the closed quadrants (**D**) was also counted to assess pharmacological effects of the treatments on the locomotion of the rats. Data are expressed as mean ± SEM. VEH: vehicle (3.3% DMSO), SON1.0: 1.0 mg/kg songorine, SON2.0: 2.0 mg/kg songorine, SON3.0: 3.0 mg/kg songorine. Asterisks mark significant effects of the given treatment compared to the VEH treatment according to post-hoc Dunnett’s test (**A**) or Dunn’s test (**B**–**D**): * *p* < 0.05.

**Figure 5 pharmaceutics-14-02067-f005:**
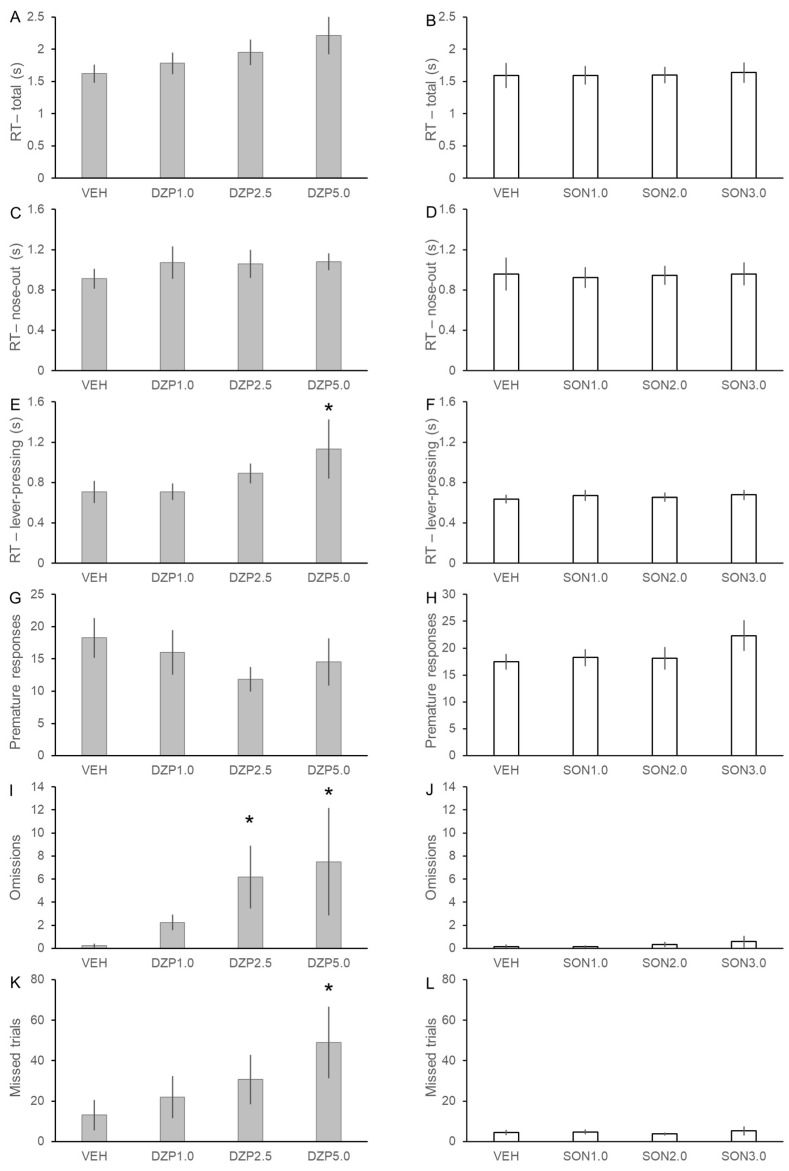
Effects of diazepam and songorine in the rat psychomotor vigilance task. Psychomotor speed was evaluated using the total reaction time (RT, panel (**A**,**B**)) and its “nose-out” (**C**,**D**) and “lever-press” (**E**,**F**) components. Overall performance was evaluated by the number of premature responses (**G**,**H**), omissions (**I**,**J**), and missed trials (**K**,**L**). Data are expressed as mean ± SEM. VEH: vehicle, DZP1.0: 1.0 mg/kg diazepam, DZP2.5: 2.5 mg/kg diazepam, DZP5.0: 5.0 mg/kg diazepam, SON0.5: 0.5 mg/kg songorine, SON1.0: 1.0 mg/kg songorine, SON2.0: 2.0 mg/kg songorine. Asterisks mark significant effects of the given treatment compared to the corresponding VEH treatment according to post-hoc LSD test: * *p* < 0.05.

**Table 1 pharmaceutics-14-02067-t001:** Summary of neuronal firing rate changes (increase: ↑, decrease: ↓, or no change: Ø) in the rat brain elicited by local administration of various pharmacological agents and their combinations. Abbreviations: GABA, gamma-aminobutyric acid; SON, songorine; PIC, picrotoxin; SAC, saclofen. Asterisks mark significant effects after the iontophoresis of the given compound based on normalized firing rates (one-sample *t*-test: * *p* < 0.05, *** *p* < 0.001).

Compound	Effect	n_neuron_	%_neuron_	n_trials_	%_trials_	Normalized Firing Rate (Mean ± S.E.M.)
**GABA**	↑	3/123	2.4%	7/296	2.4%	0.604 ± 0.049 ***
	Ø	27/123	22.0%	36/296	12.2%	
	↓	93/123	75.6%	253/296	85.5%	
**SON**	↑	1/112	0.9%	2/201	1.0%	0.621 ± 0.041 ***
	Ø	21/112	18.8%	36/201	17.9%	
	↓	90/112	80.4%	163/201	81.1%	
**PIC**	↑	24/56	42.9%	39/90	43.3%	1.150 ± 0.091
	Ø	32/56	57.1%	51/90	56.7%	
	↓	-	-	-	-	
**SAC**	↑	11/49	22.5%	20/75	26.7%	1.069 ± 0.042
	Ø	35/49	71.4%	48/75	64.0%	
	↓	3/49	6.1%	7/75	9.3%	
**SON + PIC**	↑	1/56	1.8%	2/99	2.0%	0.915 ± 0.034 *
	Ø	42/56	75.0%	68/99	68.7%	
	↓	13/56	23.2%	29/99	29.3%	
**SON + SAC**	↑	2/37	5.4%	5/89	5.6%	0.617 ± 0.089 ***
	Ø	5/37	13.5%	13/89	14.6%	
	↓	30/37	81.1%	71/89	79.8%	

## Data Availability

Data available on request from the authors.

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
