# Peer review of "Aconitum Alkaloid Songorine Exerts Potent Gamma-Aminobutyric Acid-A Receptor Agonist Action In Vivo and Effectively Decreases Anxiety without Adverse Sedative or Psychomotor Effects in the Rat"

_pharmaceutics, 2022, doi:10.3390/pharmaceutics14102067_

Round 1
Reviewer 1 Report
This manuscript describes the pharmacological and functional characterisation of a diterpenoid alkaloid songorine isolated from the geneus Aconitum. It was shown to be antagonist at GABA type A receptors through in vivo electrophysiological recordings. The findings in this study are in contrast with the previous findings where songorine is antagonist at GABA type A receptors. The clue launching the work was the link between songorine application and GABA type A receptors in CNS, explaining the pathology of anxiety-like behavior in rats. Overall, the manuscript is well written and contained substantial information supporting the findings reported. Songorine and its derivatives have many pharmacological effects such as anti-arrhythmic, anti-cardiac-fibrillation, anxiolytic effects, anti-nociceptive, anti-inflammatory, and a regenerative effect in a skin excision wound animal model. Despite its pharmacotherapeutic potential, I think the authors should address and discuss why songorine has never been tested in clinical trials. Moreover, there is little consideration of future directions. There are no doubt many more possible questions or perspectives worth considering. The authors could use this unique opportunity to provide reasoned speculations in areas of particular interest, new insights in the field, and propose possible solutions to current challenges in the clinic.
Author Response
We are grateful to the Reviewer for supporting the publication of our study and for the valuable comments on the MS. According to the suggestions, we added a paragraph to the Discussion section on the lack of clinical trials with SON and the potential future directions of research in the field (pg. 13, ln. 500-507):
„Although the effectiveness of SON for several conditions has been shown in preclini-cal studies [3], no clinical trials have investigated its potential benefits in human therapy, so far. Obviously, the body of evidence on the pharmacokinetic and pharmacodynamic properties, as well as on the mechanisms of action of SON are still not sufficiently large to step further towards testing on humans. Hopefully, our present report and other relevant findings will promote further initiatives aiming at the better understanding of the phar-macological action of SON, the identification of cellular targets, and possibly the design of several lead molecules by the modification of SON for further drug development.”
Reviewer 2 Report
I have read the manuscript entitled “Aconitum alkaloid songorine exerts potent gamma-aminobutyric acid A receptor agonist action in vivo and effectively decreases anxiety without adverse sedative or psychomotor effects in rats.” The authors in this manuscript show the potential anxiolytic activity of songorine in rats. This activity as suggested by the authors is related to the GABA-A receptor agonist properties of songorine. Although the authors have taken an interesting topic (there is still a need to find new anxiolytics without sedative component) I have to notice some problems, and these are my comments for the authors’ consideration:
1. I recognize some mistake in the age of the rats used in these studies. The authors declare the investigation on 'adult male Long Evans rats” (pp.2 ll.79 –without exactly description of rats age) while further (pp. 4 ll166) the authors declare the age of rats as '6-7 months old', the authors also state that there are adult rats is not true. Data from the literature show that a rat at 2 months of age (60 days) is considered an adult, and the social maturity of rats occurs between the ages of 5 and 6 months. After 6–7 months, rats begin to show signs of ageing (Sengupta, 2013; Andreollo et al. 2012; Quin, 2055). The age of rats used in this study is important because age-related differences in anxiety level were shown (Ferguson and Gray, 2005).
2. I recognize the next mistake: The authors declare the usage of 33% DMSO as a solvent for songorine (pp. 3 ll. 111 -112) in in vivo electrophysiological studies. In my opinion, such a high concentration of DMSO should not be used in vivo studies. However, the authors declare the lack of impact of DMSO on the neuronal firing, but it is hard to believe that giving a 33% solution of DMSO to the brain does not impact tissue wellbeing. Especially, additional behavioral studies in which songoroine was administered subcutaneously were carried out with 3.3% DMSO solution as a solvent for songorine. Why were different concentrations of DMSO used in electrophysiological and behavioral studies?
3. I also recognize some troubles with the methodology of the usage of the reference compound – dizazepam. First, why was diazepam tested in the EOM 30 min after s.c. injection and the second question why so high doses of diazepam were used? The literature data show strong sedative and locomotor impaired activity of diazepam given s.c. and i.p. 30 min before the test, its anxiolytic-like activity may be observed 60 min after s.c. or i.p. injection, because 1 hour after this route of administration the strong sedative activity of diazepam decreases while the anxiolytic-like activity is still present (Frankowska et al. 2007; Bell et al. 2013). Hence in behavioral tests such as elevated plus maze or elevated zero maze, the diazepam should be investigated 60 min after s.c. or i.p. administration to avoid any adulteration connected with its properties, i.e. locomotor impairment. My next question: Why was songorine injected in a volume of 2 ml/kg while diazepam was administered in a volume of 1ml/kg in the same route (s.c.) of administration? There is no information about the volume of vehicle.
4. Why were different numbers of animals used for the reference compound (n = 12) versus the songorine group (n=9)?
5. The Authors tested rats in the Open field test before the EOM. Please specify when exactly the test has been done before the EOM, 'a few days before” (pp. 4 ll.147) is not enough. Moreover, the authors do not show any results of OF as well as do not describe how many animals were excluded after the OF test from the EOM experiment.
6. Lack of the description of statistical methods used in the psychomotor vigilance task.
7. The method parts of MS should be rewritten. The present form MS is very hard to read, a lot of information is repeated, while some important information is missed. The statistics should be described in one place as songorine, as well as the animals and drugs used in the study.
8. The discussion part of MS should be expanded. The authors postulated the anxiolytic-like activity of songorine connected with its GABA-A agonistic properties, but this compound may act due to many other targets (e.g. its dopaminergic properties mentioned by Authors) and neurotransmitters. The tests used to assess the anxiolytic-like activity (EOM) or psychomotor activity (PVT) are not specific only for GABA-A receptor ligand or modulators.
Author Response
We are grateful to the Reviewer for supporting the publication of our study and for the valuable comments on the MS. Please, see our responses to the queries below.
Point 1: "I recognize some mistake in the age of the rats used in these studies. The authors declare the investigation on 'adult male Long Evans rats” (pp.2 ll.79 –without exactly description of rats age) while further (pp. 4 ll166) the authors declare the age of rats as '6-7 months old', the authors also state that there are adult rats is not true. Data from the literature show that a rat at 2 months of age (60 days) is considered an adult, and the social maturity of rats occurs between the ages of 5 and 6 months. After 6–7 months, rats begin to show signs of ageing (Sengupta, 2013; Andreollo et al. 2012; Quin, 2055). The age of rats used in this study is important because age-related differences in anxiety level were shown (Ferguson and Gray, 2005)."
Response1: Thank you for your comment. Indeed, age of rats in behavioral studies is an important factor. However, neither our own experience nor the literature references provided by the Reviewer confirm that 6-7 months old rats should be considered as aged animals. We have been working with aged rats in cognitive behavioral experiments for many years now. In our cognitive tasks, rats start to show systemic behavioral deficits from about 27 months of age. Thus, we usually consider rats as aged from at least 24 months of age.
These own observations are confirmed by Andreollo et al. 2012, where it is stated that 6 months of age of a rat is equal to about 18 years of human age, while a 24 months old rat should be considered as similarly aged as a 60 years human. Furthermore, the study of Ferguson and Gray (2005) mentioned by the Reviewer, found changes in the elevated plus maze performance of rats at 11 months of age. Thus, this age is much higher than the age of the animals in our study, so it does not suggest that we need to consider effects of aging on the results of our EOM experiments. Furthermore, Ferguson and Gray (2005) used rat strains that are substantially different from rats used in our experiments (Wistar-Kyoto and Sprague-Dawley rats vs. Long Evans). The referenced publication was about albino rats which are more sensitive to light, stress and anxiety than hooded (pigmented) rats, while Wistar-Kyoto rats also display depression-like behavior.
Overall, in the presently reported experiments, the 6-7 months old rats should be considered as young adults, which age is highly relevant in the context of investigating pharmacological effects on anxiety.
Andreollo NA, Santos EF, Araújo MR, Lopes LR. Rat's age versus human's age: what is the relationship? Arq Bras Cir Dig. 2012 Jan-Mar;25(1):49-51. English, Portuguese. doi: 10.1590/s0102-67202012000100011. PMID: 22569979.
Ferguson SA, Gray EP. Aging effects on elevated plus maze behavior in spontaneously hypertensive, Wistar-Kyoto and Sprague-Dawley male and female rats. Physiol Behav. 2005 Aug 7;85(5):621-8. doi: 10.1016/j.physbeh.2005.06.009. PMID: 16043200.
Point 2: "I recognize the next mistake: The authors declare the usage of 33% DMSO as a solvent for songorine (pp. 3 ll. 111 -112) in in vivo electrophysiological studies. In my opinion, such a high concentration of DMSO should not be used in vivo studies. However, the authors declare the lack of impact of DMSO on the neuronal firing, but it is hard to believe that giving a 33% solution of DMSO to the brain does not impact tissue wellbeing. Especially, additional behavioral studies in which songoroine was administered subcutaneously were carried out with 3.3% DMSO solution as a solvent for songorine. Why were different concentrations of DMSO used in electrophysiological and behavioral studies?"
Response 2: Thank you for noticing the difference in the concentration of DMSO in electrophysiological and behavioral experiments. The reason of the difference is in the technical characteristics of microiontophoretic delivery. Microiontophoresis aims the controllable delivery of bioactive compounds in small amounts in the close vicinity of the recorded neurons [1]. This is achieved via the narrow micropipettes (a few microns in diameter) surrounding the recording electrode. To ensure sufficient conductivity for the ejection current, the solution should contain the bioactive compound in a high concentration. In the case of SON, 16 mM pipette concentration was used. Because of the poor solubility of SON in water, we had to use 33% DMSO to make sufficiently concentrated SON solutions for microiontophoresis. Note, that pipette concentration is not equal to the resulting drug concentration in the brain tissue, as the latter is a function of the applied ejection current [2–4]. As in the solution only SON was a polarized ingredient, during microiontophoretic delivery, mainly SON molecules were supposed to be ejected by the electric current from the capillary and to enter the brain tissue. Although a little electroosmotic flow is also generated parallel to drug delivery [5], this may only carry a small amount of DMSO into the brain. The lack of effect of such a small amount of DMSO, delivered through microiontophoresis in the vicinity of cells was also reported in our recent paper [6]. Most importantly, control measurements in the present study also fully confirmed that the DMSO content of the SON solution used for microiontophoresis did not influence the electrophysiological results (see p. 3, l. 124-125).
1. Gottschaldt, K.M.; Hicks, T.P.; Vahle-Hinz, C. A Combined Recording and Microiontophoresis Technique for Input-Output Analysis of Single Neurons in the Mammalian CNS. J. Neurosci. Methods 1988, 23, 233–239.
2. Kirkpatrick, D.C.; Walton, L.R.; Edwards, M.A.; Wightman, R.M. Quantitative Analysis of Iontophoretic Drug Delivery from Micropipettes. Analyst 2016, 141, 1930–1938, doi:10.1039/c5an02530c.
3. Kirkpatrick, D.C.; Mark Wightman, R. Evaluation of Drug Concentrations Delivered by Microiontophoresis. Anal. Chem. 2016, 88, 6492–6499, doi:10.1021/acs.analchem.6b01211.
4. Gerhardt, G.A.; Palmer, M.R. Characterization of the Techniques of Pressure Ejection and Microiontophoresis Using in Vivo Electrochemistry. J. Neurosci. Methods 1987, 22, 147–159.
5. Herr, N.R.; Kile, B.M.; Carelli, R.M.; Wightman, R.M. Electroosmotic Flow and Its Contribution to Iontophoretic Delivery. Anal Chem 2008, 80, 8635–8641.
6. Nagy, L.V.; Bali, Z.K.; Kapus, G.; Pelsőczi, P.; Farkas, B.; Lendvai, B.; Lévay, G.; Hernádi, I. Converging Evidence on D-Amino Acid Oxidase–Dependent Enhancement of Hippocampal Firing Activity and Passive Avoidance Learning in Rats. Int. J. Neuropsychopharmacol. 2021, 24, 434–445, doi:10.1093/ijnp/pyaa095.
Point 3: "I also recognize some troubles with the methodology of the usage of the reference compound – dizazepam. First, why was diazepam tested in the EOM 30 min after s.c. injection and the second question why so high doses of diazepam were used? The literature data show strong sedative and locomotor impaired activity of diazepam given s.c. and i.p. 30 min before the test, its anxiolytic-like activity may be observed 60 min after s.c. or i.p. injection, because 1 hour after this route of administration the strong sedative activity of diazepam decreases while the anxiolytic-like activity is still present (Frankowska et al. 2007; Bell et al. 2013). Hence in behavioral tests such as elevated plus maze or elevated zero maze, the diazepam should be investigated 60 min after s.c. or i.p. administration to avoid any adulteration connected with its properties, i.e. locomotor impairment. My next question: Why was songorine injected in a volume of 2 ml/kg while diazepam was administered in a volume of 1ml/kg in the same route (s.c.) of administration? There is no information about the volume of vehicle."
Response 3: Thank you for your remarks about the administration of diazepam. Indeed, many studies reported anxiolytic effect of diazepam at lower doses than that we used. Furthermore, in some studies the drug was applied earlier than 30 min prior to behavioral testing. However, we planned our experiments according to literature data, where diazepam was applied in a similar manner (dose and pretreatment time) as in our study. We are citing some examples below.
• Harro J, Põld M, Vasar E. Anxiogenic-like action of caerulein, a CCK-8 receptor agonist, in the mouse: influence of acute and subchronic diazepam treatment. Naunyn Schmiedebergs Arch Pharmacol. 1990 Jan-Feb;341(1-2):62-7. doi: 10.1007/BF00195059. PMID: 2314484.
• Harro J, Lang A, Vasar E. Long-term diazepam treatment produces changes in cholecystokinin receptor binding in rat brain. Eur J Pharmacol. 1990 May 3;180(1):77-83. doi: 10.1016/0014-2999(90)90594-v. PMID: 2365005.
• Saleem S, Naqvi F, Batool A, Naqvi SH, Naqvi F, Batool Z, Tabassum S, Haider S. Neuroprotective role of a monoterpene (thymol) on diazepam induced withdrawal symptoms in rats. Pak J Pharm Sci. 2021 Jul;34(4(Supplementary)): 1615-1620. PMID: 34799339.
• Kniazev GG, Nikiforov AF, Tolochko ZS, Mikhaĭlov VV. Vliianie seduksena na uslovnoreflektornoe povedenie i reaktivnost' noradrenergicheskoĭ sistemy mozga krys [Effect of seduxen on conditioned reflex behavior and reactivity of the noradrenergic system of the rat brain]. Farmakol Toksikol. 1983 Nov-Dec;46(6):15-7. Russian. PMID: 6653751.
• Malyshev AV, Sukhanova IA, Ushakova VM, Zorkina YA, Abramova OV, Morozova AY, Zubkov EA, Mitkin NA, Pavshintsev VV, Doronin II, Gedzun VR, Babkin GA, Sanchez SA, Baker MD, Haile CN. Peptide LCGA-17 Attenuates Behavioral and Neurochemical Deficits in Rodent Models of PTSD and Depression. Pharmaceuticals (Basel). 2022 Apr 12;15(4):462. doi: 10.3390/ph15040462. PMID: 35455459; PMCID: PMC9029485.
• Matto V, Harro J, Allikmets L. The effects of cholecystokinin A and B receptor antagonists on exploratory behaviour in the elevated zero-maze in rat. Neuropharmacology. 1997 Mar;36(3):389-96. doi: 10.1016/s0028-3908(97)00011-7. PMID: 9175618.
• Hawiset T, Sriraksa N, Kamsrijai U, Wanchai K, Inkaew P. Anxiolytic and antidepressant-like activities of aqueous extract of Azadirachta indica A. Juss. flower in the stressed rats. Heliyon. 2022 Feb 3;8(2):e08881. doi: 10.1016/j.heliyon.2022.e08881. PMID: 35198760; PMCID: PMC8844689.
• Motaghi S, Moghaddam Dizaj Herik H, Sepehri G, Abbasnejad M, Esmaeli-Mahani S. The anxiolytic effect of salicylic acid is mediated via the GABAergic system in the fear potentiated plus maze behavior in rats. Mol Biol Rep. 2022 Feb;49(2):1133-1139. doi: 10.1007/s11033-021-06939-0. Epub 2021 Nov 19. PMID: 34797490.
We think that it is normal that such details of drug administration varies between laboratories because many circumstances influence the optimal application of a certain drug. For example, different formulations of the same active agent may possess different pharmacokinetic properties. In our experiments, we used a commercially available formulation of diazepam (Seduxen, Gedeon Richter Plc.) which is formulated in a specific vehicle that was described in the Methods section (p. 4, l. 183-185). Perhaps, this difference in the vehicle may be responsible of different absorption properties and thus, differences in optimal dosing and timing.
Furthermore, as we were also interested in the locomotor side-effects of SON vs. DZP, we do not find it an issue that DZP was applied in a manner that also exerts these locomotor effects parallel to its anxiolytic effect.
We usually aim to apply the lowest possible volume of drugs during subcutaneous administrations. However, the solubility of DZP and SON is different, thus, we needed to apply SON in a lower concentration and consequently in a higher volume. The volume of administration was further clarified also in the Psychomotor vigilance task section of the revised MS (p. 5, l. 247).
The volume of the vehicle was the same as the volume of the test compound in a certain experiment. Now, we clarified this in the MS, and can be found at p. 4, l. 186 and p. 4, l. 193.
Point 4: "Why were different numbers of animals used for the reference compound (n = 12) versus the songorine group (n=9)?"
Response 4: During the planning of the EOM experiments it was supposed that in the DZP experiment, more animals will be excluded because of lack of activity. Therefore, it was reasonable to start DZP experiments with a higher initial number of animals. Although SON was tested on less animals than DZP, statistical power was still enough to successfully demonstrate the anxiolytic effect of the compound.
Point 5: "The Authors tested rats in the Open field test before the EOM. Please specify when exactly the test has been done before the EOM, 'a few days before” (pp. 4 ll.147) is not enough. Moreover, the authors do not show any results of OF as well as do not describe how many animals were excluded after the OF test from the EOM experiment."
Response 5: According to the Reviewer’s comment, we specified in the MS that the OF measurements were made 2-6 days before the EOM experiment. (p. 4, l. 151)
We did not report the results of the OF experiments, because there were no experimental manipulations, and it was only run for the purpose of prior phenotyping to generate treatment groups with uniform distribution of the basic behavioral traits of the animals. We think that these results lack scientific interest, therefore, we decided not include them in the MS. However, the data of OF experiments are available upon request.
We further specified in the MS the number of outliers in the OF test. „Altogether 4 animals were excluded from EOM experiment because of outlier performance in the OF test” (p. 4 l. 157-159).
Point 6: "Lack of the description of statistical methods used in the psychomotor vigilance task."
Response 6: Thank you for your comment, we added the description of statistical analysis to the Psychomotor vigilance task section (p. 6, l. 261-265).
Point 7: "The method parts of MS should be rewritten. The present form MS is very hard to read, a lot of information is repeated, while some important information is missed. The statistics should be described in one place as songorine, as well as the animals and drugs used in the study."
Response 7: Thank you for suggestions. In the revised MS, we restructured the Methods section to shorter subsections that may aid better readability. Furthermore, we pursued to simplify some detailed technical description, and we included information that were earlier missed from the MS.
Point 8: "The discussion part of MS should be expanded. The authors postulated the anxiolytic-like activity of songorine connected with its GABA-A agonistic properties, but this compound may act due to many other targets (e.g. its dopaminergic properties mentioned by Authors) and neurotransmitters. The tests used to assess the anxiolytic-like activity (EOM) or psychomotor activity (PVT) are not specific only for GABA-A receptor ligand or modulators."
Response 8: Thank you for your suggestions to expand the scope of the Discussion. We modified the part of the Discussion that describes the possible targets of SON other than GABA(A) receptors. We elaborated on the interpretation of earlier studies that had suggested the potential dopaminergic effects of SON, and we emphasized the possible connections between the available prior electrophysiological data and our present behavioral results. Furthermore, we emphasized the limitations of EOM and PVT tests to clarify the interpretations of neurotransmitter selective effects.
Round 2
Reviewer 2 Report
I have read the revised version of the manuscript entitled “Aconitum alkaloid songorine exerts potent gamma-aminobutyric acid A receptor agonist action in vivo and effectively decreases anxiety without adverse sedative or psychomotor effects in rats.” I still recognize some problems and these are my comments for the consideration of the authors.
1. Why were different sources of rats used for electrophysiological studies (Charles River Laboratories, Gödöll, 93 Hungary ) vs behavioral studies (Janvier Labs, Le Genest-Saint-Isle, France)?
2. I still do not understand why for electropisilogical studies in vivo 33% DMSO was used as a solvent for songorine when for behavioral studies the authors used also dissolved songorine but in a solution of 3,3% DMSO and phosphate buffer. Furthermore, the 6 data of the literature cited by the authors in the response as an explanation for the usage of such a high concentration of DMSO in in vivo electrophysiological studies in only one work (Nagy et al. 2021) 40% DMSO was used, but in the other articles (different labs and authors) the artificial cerebral spinal fluid (aCSF) consisted of 126 mM NaCl, 2.5 mM KCl, 1 mM NaH2PO4, 26 mM NaHCO3, 2 mM MgSO4, 2 mM CaCl2 and 1 mM glucose and adjusted to pH 7.4. aCSF is highly recommended as a vehicle and solvent for compounds that are given into the brain, in electrophysiological studies, as well.
3. I also recognize similar trouble with the explanation of the dose of diazepam and time of injection prior to EOM. The authors' explanations are insufficient and the literature cited in support of their diazepam administration schedule is not adequate as its concerns mice (Harro et al 1990) or diazepam was given in the repeated scheme of administration (Harro et al, 1990 – two-week treatment; Saleem et al. 2021 – 14-day treatment; Hawiset et al 2022 – 30-day treatment). Only two works cited by the authors confirmed anxiolytic activity of diazepam 30 min after injection of ip without sedative activity and locomotor impairment. I agree with the arguments of the authors regarding the different pharmacokinetic parameters of the commercially available diazepam formulation, but I disagree with the use of another explanation in which the different diazepam treatment scheme was used.
Author Response
We appreciate the further queries of the Reviewer.
Point 1: Why were different sources of rats used for electrophysiological studies (Charles River Laboratories, Gödöll, 93 Hungary ) vs behavioral studies (Janvier Labs, Le Genest-Saint-Isle, France)?
Response 1: Thank you for your question. The reason is that Charles River Laboratories changed their distributor in Hungary, however, the service of the new distributor was not satisfactory, and we had to change supplier. Apart from that, we used the same outbred strain (Long Evans) in the study. Also, the animal cohort for the same experiment was purchased in one batch from the same supplier to ensure the same genetic background of the animals. We believe that using the same strain and changing suppliers between experiments (and not within) should not have influenced the quality of the scientific results obtained.
Point 2: I still do not understand why for electropisilogical studies in vivo 33% DMSO was used as a solvent for songorine when for behavioral studies the authors used also dissolved songorine but in a solution of 3,3% DMSO and phosphate buffer. Furthermore, the 6 data of the literature cited by the authors in the response as an explanation for the usage of such a high concentration of DMSO in in vivo electrophysiological studies in only one work (Nagy et al. 2021) 40% DMSO was used, but in the other articles (different labs and authors) the artificial cerebral spinal fluid (aCSF) consisted of 126 mM NaCl, 2.5 mM KCl, 1 mM NaH2PO4, 26 mM NaHCO3, 2 mM MgSO4, 2 mM CaCl2 and 1 mM glucose and adjusted to pH 7.4. aCSF is highly recommended as a vehicle and solvent for compounds that are given into the brain, in electrophysiological studies, as well.
Response 2: Thank you for your further comment on the validity of using higher concentrations of DMSO in microiontophoresis. Perhaps, our previous response was ambiguous concerning the cited references. The references that were cited about the microiontophoretic technique were not all intending to prove the inertness of DMSO in the brain, but rather to explain the major hallmarks and basic technical requirements of the mictoiontophoretic technique. Namely, for iontophoresis, it is required to use high pipette concentrations of the bioactive compounds to ensure sufficient electric conductivity for the delivery of the electrically charged molecules/ions with constant current applied in the range of 1 to 500 nA per channel. However, as the Reviewer also pointed out, we cited a previous article to support the fact that microiontophoretically applied DMSO does not exert any observable effects on the electrophysiological properties of the recorded neurons neither in the present (where it was applied as control), nor in a previous study. This fact may be due to the non-polarized (inert) properties of DMSO and its inability to be forced to move by constant electric fields. Although aCSF may be an optimal vehicle due to its composition, it is a highly polarized water-based buffer that is not appropriate to hold compounds with poor water-solubility in a stable solution. Furthermore, for microiontophoresis, it is a basic requirement to apply vehicles that are electrically inert (not charged) or at least much less polarized, since otherwise the charged particles with small molecular weight and large mobility (e.g., sodium-ions, bicarbonate-ions, phosphate-ions, etc.,) will only be delivered and not the pharmacologically active and much larger particles (such as songorine molecules in our case). Again, we would like to emphasize the substantial difference between the microiontophoretic and microinjection techniques in the driving force of delivery, which is the electrostatic field induced by the applied constant current in the case of microiontophoresis, and not the pressure that is exerted on the syringe pump in the case of microinjection. Thus, using microiontophoresis, the electrically (charged) polarized molecules (e.g., songorine) should be selectively ejected by the electrostatic field, while the neutral molecules such as the solute (DMSO) are not forced to move into the tissue but should remain in the pipette.
As we described previously, in our case, the electrically neutral DMSO could be delivered only in a very small (negligible) quantity by the electroosmotic flow during the ejection of the pharmacologically active compound (songorine), and that amount was very unlikely to result in any observable effect in the extracellular space (indeed, we did not record such effects neither in the present nor in previous studies).
In summary, to the best of our knowledge, there are no known theoretical and practical reasons to question the current methodology used for the iontophoretic application of songorine. The experiments were also properly controlled by confirming that the microiontophoretic application of the vehicle did not influence any observable electrophysiological properties of the recorded neurons, which we also state in the revised MS (p. 3, l. 116-117).
Please kindly refer to the following methodological papers on the microiontophoresis technique:
1. Gerhardt, G.A.; Palmer, M.R. Characterization of the Techniques of Pressure Ejection and Microiontophoresis Using in Vivo Electrochemistry. J. Neurosci. Methods 1987, 22, 147–159.
2. Kirkpatrick, D.C.; Mark Wightman, R. Evaluation of Drug Concentrations Delivered by Microiontophoresis. Anal. Chem. 2016, 88, 6492–6499, doi:10.1021/acs.analchem.6b01211.
3. Takmakov, P.; McKinney, C.J.; Carelli, R.M.; Wightman, R.M. Instrumentation for Fast-Scan Cyclic Voltammetry Combined with Electrophysiology for Behavioral Experiments in Freely Moving Animals. Rev Sci Instrum 2011, 82, 74302, doi:10.1063/1.3610651.
4. Gottschaldt, K.M.; Hicks, T.P.; Vahle-Hinz, C. A Combined Recording and Microiontophoresis Technique for Input-Output Analysis of Single Neurons in the Mammalian CNS. J. Neurosci. Methods 1988, 23, 233–239.
5. Purves, R.D. The Physics of Iontophoretic Pipettes. J. Neurosci. Methods 1979, 1, 165–178, doi:10.1016/0165-0270(79)90014-1.
6. Herr, N.R.; Kile, B.M.; Carelli, R.M.; Wightman, R.M. Electroosmotic Flow and Its Contribution to Iontophoretic Delivery. Anal Chem 2008, 80, 8635–8641.
7. Engberg, I.; Flatman, J. a; Lambert, J.D. Extracellular and Intracellular Recording during Micro-Iontophoresis: An Appraisal [Proceedings]. Br. J. Pharmacol. 1978, 64, 451P-452P.
8. Murnick, J.G.; Dubé, G.; Krupa, B.; Liu, G. High-Resolution Iontophoresis for Single-Synapse Stimulation. J. Neurosci. Methods 2002, 116, 65–75.
Furthermore, the following website of Kation Scientific Ltd. (the supplier of microiontophoretic equipments and electrodes used in the present study) for further technical details, manuals and references:
www.kationscientific.com
Point 3: I also recognize similar trouble with the explanation of the dose of diazepam and time of injection prior to EOM. The authors' explanations are insufficient and the literature cited in support of their diazepam administration schedule is not adequate as its concerns mice (Harro et al 1990) or diazepam was given in the repeated scheme of administration (Harro et al, 1990 – two-week treatment; Saleem et al. 2021 – 14-day treatment; Hawiset et al 2022 – 30-day treatment). Only two works cited by the authors confirmed anxiolytic activity of diazepam 30 min after injection of ip without sedative activity and locomotor impairment. I agree with the arguments of the authors regarding the different pharmacokinetic parameters of the commercially available diazepam formulation, but I disagree with the use of another explanation in which the different diazepam treatment scheme was used.
Response 3: We appreciate the comments of the Reviewer, however, we believe that the applied treatment regimen was appropriate, which is confirmed by the cited publications. Indeed, we did not aim to avoid the sedative side-effects of diazepam, since this is an important property of benzodiazepine anxiolytics, even if they mostly occur at early stages after administration. We agree with the Reviewer, that some references we cited earlier differed from our study in some aspects, however, in all of the cited papers diazepam was applied at similar doses and/or at similar pre-treatment times compared to our present study. In two of the cited papers, researchers applied the same treatment regimen that we used in our behavioural experiments. These two studies confirm the validity of the applied treatment regimen, which corresponded to the suggested therapeutical dose of diazepam for rats (0.5 to 5.0 mg/kg b.w.). Unfortunately, we have not found references in the literature for the time-course of sedative and anxiolytic effects of diazepam in rats, thus, we were unable to discuss this issue in the manuscript. Indeed, the study of Frankowska et al. (2007) (cited by the Reviewer) used a treatment regimen similar to our present experiments: i.e., diazepam was injected at 2 mg/kg b.w., and at 30 min before the behavioural tests started. Therefore, we hope that the cited references will be sufficient and acceptable for the Reviewer to indicate the validity of the chosen treatment regimen in our present study.
1. Frankowska, M.; Filip, M.; Przegaliński, E. Effects of GABAB Receptor Ligands in Animal Tests of Depression and Anxiety. Pharmacol. Reports 2007, 59, 645–655.